# Multimodal Fusion with Relational Learning for Molecular Property Prediction

## Abstract

Graph-based molecular representation learning is essential for accurately predicting molecular properties in drug discovery and materials science; however, it faces significant challenges due to the intricate relationships among molecules and the limited chemical knowledge utilized during training. While contrastive learning is often employed to handle molecular relationships, its reliance on binary metrics is insufficient for capturing the complexity of these interactions. Multimodal fusion has gained attention for property reasoning, but previous work has explored only a limited range of modalities, and the optimal stages for fusing different modalities in molecular property tasks remain underexplored. In this paper, we introduce MMFRL (Multimodal Fusion with Relational Learning for Molecular Property Prediction), a novel framework designed to overcome these limitations. Our method enhances embedding initialization through multi-modal pre-training using relational learning. We also conduct a systematic investigation into the impact of modality fusion at different stages—early, intermediate, and late—highlighting their advantages and shortcomings. Extensive experiments on MoleculeNet benchmarks demonstrate that MMFRL significantly outperforms existing methods. Furthermore, MMFRL enables task-specific optimizations. Additionally, the explainability of MMFRL provides valuable chemical insights, emphasizing its potential to enhance real-world drug discovery applications.

## 1 Introduction

Graph representation learning for molecules has gained significant attention in drug discovery and materials science, as it effectively encapsulates molecular structures and enables the effective investigation of structure-activity relationships (Wieder et al., 2020; Zhang et al., 2022; Fang et al., 2022; Wang et al., 2023). In this paradigm, atoms are treated as nodes and chemical bonds as edges, effectively encapsulating the connectivities that define molecular behavior. However, it poses significant challenges due to intricate relationships among molecules and the limited chemical knowledge utilized during training.

Often, contrastive learning (CL) is employed to study relationships among molecules, but it relies on binary metrics of positive and negative pairs, and tends to oversimplify complex molecular interactions. For example, consider Thalidomide: while the *(R)*- and *(S)*-enantiomers share the same topological graph and differ only at a single chiral center, their biological activities are drastically different—the *(R)*-enantiomer is effective in treating morning sickness, whereas the *(S)*-enantiomer causes severe birth defects. In other words, the *(R)*- and *(S)*-enantiomers are similar in terms of topological stucture but dissimilar in terms of biological activities. Thus, a more sophisticated approach is required to tackle these scenarios. A potential solution would be to use continuous metrics within a multi-view space, enabling a more comprehensive understanding of these complex molecular relationships.

When it comes to multimodal learning for molecules, we often encounter data availability and incompleteness issues. This raises a critical question: how can multimodal information be effectively leveraged for molecular property reasoning when such data is absent in downstream tasks? Recent studies have demonstrated the effectiveness of pretraining molecular Graph Neural Networks (GNNs) by integrating additional knowledge sources (Wang et al., 2021; 2022b; Liu et al., 2022a; Xu et al., 2023a). Building on this foundation, a promising solution is to pretrain multiple replicas of molecular

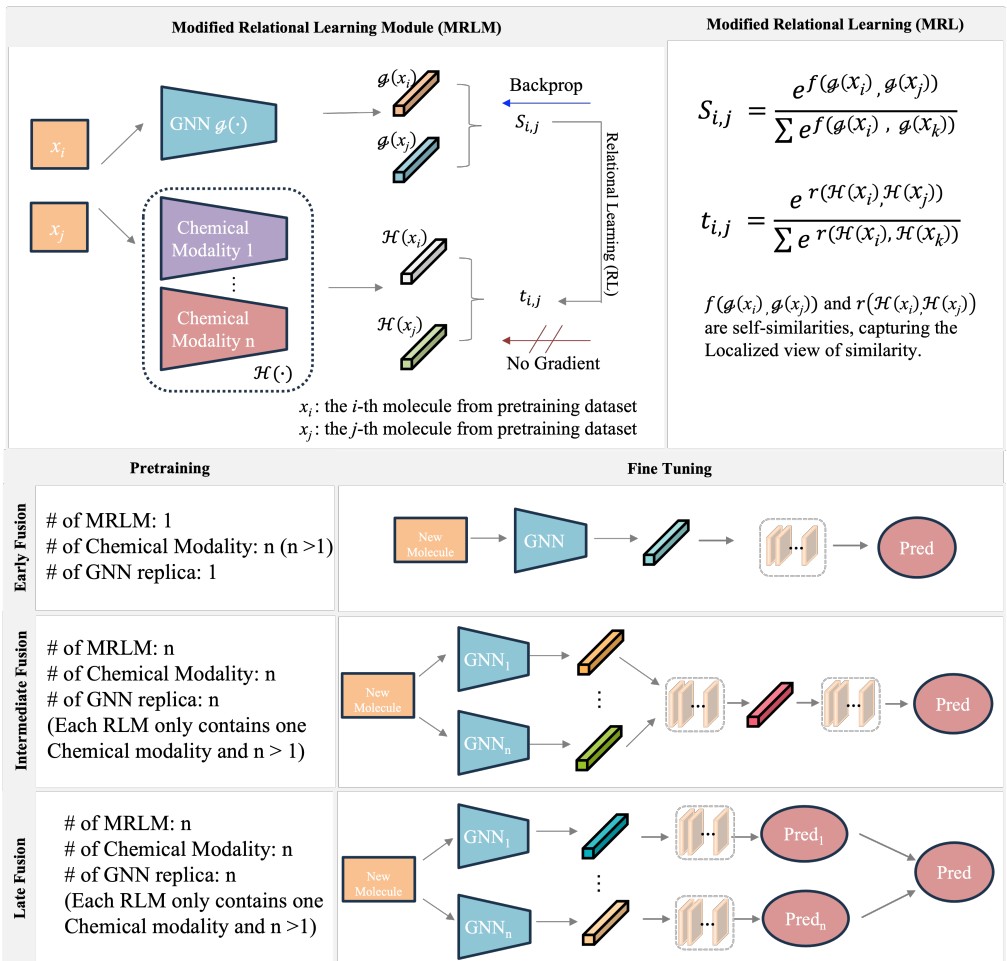

Figure 1: **Multimodal Fusion with Relational Learning for Molecular Property Prediction (MMFRL).** This figure shows our proposed idea about how to transfer the knowledge from other modalities and use fusion to improve the performance further. Unlike the general contrastive learning framework shown in Appendix Figure A.2, MMFRL doesn't need to define positive or negative pairs and is capable of learning continuous ordering from target similarity.

GNNs, with each replica dedicated to learning from a specific modality. This approach allows downstream tasks to benefit from multimodal data that is not accessible during fine-tuning, ultimately improving representation learning.

Facing these challenges and opportunities, we propose MMFRL (Multimodal Fusion with Relational Learning for Molecular Property Prediction), a novel framework features relational learning (RL) and multimodal fusion (MMF). RL utilizes a continuous relation metric to evaluate relationships among instances in the feature space (Balcan & Blum, 2006; Wen et al., 2023). Our major contribution comprises three aspects: **_Conceptually:_** We introduce a modified relational learning metric for molecular graph representation that offers a more comprehensive and continuous perspective on inter-instance relations, effectively capturing both localized and global relationships among instances. To the best of our knowledge, this is the first work to demonstrate such generalized relational learning metric for molecular graph representation. **_Methodologically:_** Our proposed modified relational metric captures complex relationships by converting pairwise self-similarity into relative similarity, which evaluates how the similarity between two elements compares to the similarity of other pairs in the dataset. In addition, we integrate these metrics into a fused multimodal representation, which has the potential to enhance performance, allowing downstream tasks to leverage modalities that are not directly accessible during fine-tuning. **_Empirically:_** MMFRL excels in various downstream tasks for

Molecular Property Predictions. Last but not least, we demonstrate the explainability of the learned representations through two post-hoc analysis. Notably, we explore minimum positive subgraphs and maximum common subgraphs to gain insights for further drug molecule design.

## 2 PRELIMINARIES

**Directed Message Passing Neural Network (DMPNN).** The Message Passing Neural Network (MPNN) (Gilmer et al., 2017) is a GNN model that processes an undirected graph $G$ with node (atom) features $x_v$ and edge (chemical bond) features $e_{vw}$. It operates through two distinct phases: a message passing phase, facilitating information transmission across the molecule to construct a neural representation, and a readout phase, utilizing the final representation to make predictions regarding properties of interest. The primary distinction between DMPNN and a generic MPNN lies in the message passing phase. While MPNN uses messages associated with nodes, DMPNN crucially differs by employing messages associated with directed edges (Yang et al., 2019). This design choice is motivated by the necessity to prevent totters (Mahé et al., 2004), eliminating messages passed along paths of the form $v_1 v_2 \ldots v_n$, where $v_i = v_{i+2}$ for some $i$, thereby eliminating unnecessary loops in the message passing trajectory.

**Relational Learning.** Original Relation Learning (Zheng et al., 2021) ensures that different augmented views of the same instance from computer vision tasks share similar features, while allowing for some variability. This approach captures the essential characteristics of the instance, promoting consistency across the views without requiring them to be identical. By doing so, it enhances the model's ability to generalize and recognize underlying patterns in the data. Suppose $z_i$ is the original embdding for the $i - th$ instance. Then $z_i^1$ is the embedding of first augmented view for $z_i$, and $z_i^2$ is the embedding of second augmented view for $z_i$. In this case, the Loss of Relational Learning (RL) is formulated as following:

$$s_{ik}^1 = \frac{1_{i \neq k} \cdot \exp(z_i^1 \cdot z_k^2 / \tau)}{\sum_{j=1}^N 1_{i \neq j} \cdot \exp(z_i^1 \cdot z_j^2 / \tau)}$$

$$s_{ik}^2 = \frac{1_{i \neq k} \cdot \exp(z_i^2 \cdot z_k^2 / \tau_m)}{\sum_{j=1}^N 1_{i \neq j} \cdot \exp(z_i^2 \cdot z_j^2 / \tau_m)}$$

$$L_{RL} = -\frac{1}{N} \sum_{i=1}^N \sum_{\substack{k=1 \\ k \neq i}}^N s_{ik}^2 \log(s_{ik}^1).$$

**Multi-Modality Fusion.** Multi-Modality Fusion combines diverse heterogeneous data (e.g. text, images, graph) to create a more comprehensive understanding of complex scenarios (Lahat et al., 2015; Khaleghi et al., 2013; Poria et al., 2015; Ramachandram & Taylor, 2017; Pawłowski et al., 2023). This approach leverages the strengths of each modality, potentially improving performance in tasks like sentiment analysis or medical diagnosis. While challenging to implement due to the need to align different data streams, successful fusion can provide insights beyond what's possible with single modalities, advancing AI and data-driven decision-making. In particular, the way to fuse different modality should also depends on the dominace of each unimodality (Pawłowski et al., 2023).

## 3 METHODS

We first explain our proposed modified metric in relational learning to facilitate smooth alignment between graph and referred unimodality. Then, we introduce approaches for integrating multi modalities at different stages of the learning process.

### 3.1 MODIFIED RELATIONAL LEARNING IN PRETRAINING

We propose a modified relational metric by adapting the softmax function as a pairwise weighting mechanism. Let $|\mathcal{S}|$ denote the size of the instance set. The variable $s_{i,j}$ represents the learned similarity distribution where $z_i$ is the embedding to be trained. On the other hand, $t_{i,j}^R$ defines the

target similarity distribution that captures the relationship between the pair of instances in the given space or modality $R$, where $z_i^R$ is a fixed embedding. The detailed formulation for the Loss of Modified Relatioal Learning (MRL) is provided below:

$$s_{i,j} = \frac{\exp(sim(z_i, z_j))}{\sum_{k=1}^{|\mathcal{S}|} \exp(sim(z_i, z_k))} \tag{1}$$

$$t_{i,j}^R = \frac{\exp(sim(z_i^R, z_j^R))}{\sum_{j=1}^{|\mathcal{S}|} \exp(sim(z_i^R, z_k^R))} \tag{2}$$

$$L_{MRL} = -\frac{1}{|\mathcal{S}|} \sum_{i=1}^{|\mathcal{S}|} \sum_{j=1}^{|\mathcal{S}|} t_{i,j}^R \log(s_{i,j}). \tag{3}$$

Notably, unlike other similarity learning approaches (Wang et al., 2019; Zhang et al., 2021), our method does not rely on the categorization of negative and positive pairs for the pair weighting function. Additionally, the use of the softmax function ensures that the generalized target similarity $t_{i,j}$ adheres to the principles of convergence as following:

**Theorem 3.1** (Convergence of Modified Relational Learning Metric). *Let $\mathcal{S}$ be a set of instances with size of $|\mathcal{S}|$, and let $\mathcal{P}$ represent the learnable latent representations of instances in $\mathcal{S}$ such that $|\mathcal{P}| = |\mathcal{S}|$. For any two instances $i, j \in \mathcal{S}$, their respective latent representations are denoted by $\mathcal{P}_i$ and $\mathcal{P}_j$. Let $t_{i,j}$ represent the target similarity between instances $i$ and $j$ in a given domain, and let $d_{i,j}$ be the similarity between $\mathcal{P}_i$ and $\mathcal{P}_j$ in the latent space. If $t_{i,j}$ is non-negative and $\{t_{i,j}\}$ satisfies the constraint $\sum_{j=1}^{|\mathcal{S}|} t_{i,j} = 1$, consider the loss function for an instance $i$ defined as follows:*

$$L(i) = -\sum_{j=1}^{|\mathcal{S}|} t_{i,j} \log \left( \frac{e^{d_{i,j}}}{\sum_{k=1}^{|\mathcal{S}|} e^{d_{i,k}}} \right) \tag{4}$$

*then when it reaches ideal optimum, the relationship between $t_{i,j}$ and $d_{i,j}$ satisfies:*

$$softmax(d_{i,j}) = t_{i,j} \tag{5}$$

For detailed proof, please refer to Appendix Section B.1.

### 3.2 FUSION OF MULTI-MODALITY INFORMATION IN DOWNSTREAM TASKS.

During pre-training, the encoders are initialized with parameters derived from distinct reference modalities. A critical question that arises is how to effectively utilize these pre-trained models during the fine-tuning stage to improve performance on downstream tasks.

#### 3.2.1 EARLY STAGE: MULTIMODAL MULTI-SIMILARITY

With a set of known target similarity $\{t^R\}$ from various modalities, we can transform themto multimodal space through a fusion function. There are numerous potential designs of the fusion function. For simplicity, we take linear combination as a demonstration. The multimodal generalized multi-similarity $t_{i,j}^M$ between $i^{th}$ and $j^{th}$ objects can be defined as follows:

$$t_{i,j}^M = fusion(\{t^R\}) \tag{6}$$

$$= \sum w_R \cdot t_{i,j}^R \tag{7}$$

where $t_{i,j}^R$ represents the target similarity between $i^{th}$ and $j^{th}$ instance in unimodal space $R$, $w_R$ is the pre-defined weights for the corresponding modal, and $\sum w_R = 1$. Then we can make $t_{i,j} = t_{i,j}^R$ in equation 3. Such that, it still satisfy the requirement of convergence (See proof in Appendix SectionB.2). In this case, the learnt similarity during pretraining will be aligned with this new combined target similarity.

### 3.2.2 INTERMEDIATE STAGE: EMBEDDING CONCATENATION AND FUSION

Intermediate fusion integrates features from various modalities after their individual encoding processes and prior to the decoding/readout stage. Let $\mathbf{f}_1, \mathbf{f}_2, \ldots, \mathbf{f}_n$ represent the feature vectors obtained from these different modalities. The resulting fused feature vector can be defined as follows:

$$\mathbf{f}_{\text{fused}} = \text{MLP}(\text{concat}(\mathbf{f}_1, \mathbf{f}_2, \ldots, \mathbf{f}_n)) \tag{8}$$

Where concat represents concatenation of the feature vectors. The fused features are then fed into a later readout function or decoder for downstrean tasks prediction or classification. The MLP (Multi-Layer Perceptron) is used to reduce the dimension to be the same as $\mathbf{f}_i$.

### 3.2.3 LATE STAGE: DECISION-LEVEL

Late fusion (or decision-level fusion) combines the outputs of models trained on different modalities after they have been processed independently. Each modality is first processed separately, and their predictions are combined at a later stage.

Let $p_1, p_2, \ldots, p_n$ be the predictions (e.g., probabilities) from different modalities. The final prediction $p_{\text{final}}$ can be computed using a weighted sum mechanism:

$$w_i = T_i(\mathbf{f}_i) \tag{9}$$

$$p_i = \text{readout}_i(\mathbf{f}_i) \tag{10}$$

$$p_{\text{final}} = \sum_{i=1}^{n} w_i p_i \tag{11}$$

Where $w_i$ are the weights assigned to each modality's prediction, and they can be adjusted based on the importance of each modality. In particular, $w_i$ is tunable during the learning process for respective downsteak tasks.

## 4 EXPERIMENTS

In this section, we begin by presenting the datasets and selected modalities. Subsequently, we showcase the results obtained from MMFRL. Finally, we demonstrate the explainability of the learned molecular representations. (Please refer to the experimental details of pre-training and fine-tuning in the Appendix Section D.)

### 4.1 DATASET

#### 4.1.1 SELECTED MODALITIES FOR TARGET SIMILARITY CALCULATION

The following modalities are used for target similarity calculation. For details on training the corresponding encoders to obtain fixed embeddings for these modalities, please refer to Appendix Section C.1.

**Fingerprint:** Fingerprints are binary vectors that represent molecular structures, capturing the presence or absence of particular substructures, fragments, or chemical features within a molecule.

**SMILES (Simplified Molecular Input Line Entry System):** SMILES offers a compact textual representation of chemical structures.

**NMR (Nuclear Magnetic Resonance):** NMR spectroscopy provides detailed insights into the chemical environment of atoms within a molecule. By analyzing the interactions of atomic nuclei with an applied magnetic field, NMR can reveal information about the structure, dynamics, and interactions of molecules, including the connectivity of atoms, functional groups, and conformational changes. In our experiments, $\text{NMR}_{\text{spectrum}}$ provides the information about the overal information of molecule while $\text{NMR}_{\text{peak}}$ provides the information about the individual atoms in the molecule.

**Image:** Images (e.g., 2D chemical structures) provide a visual representation of molecular structures.

Table 1: Study on the performances of MMFRL$_{Unimodality}$. The best results are denoted in bold, and the second-best are indicated with underlining among the five modalities. The first 8 tasks are for classification under evaluation of ROC-AUC, while the last three are for regression with evaluation of RMSE.

| DATA SET | BBBP | BACE | SIDER | CLINTOX | HIV | MUV | TOX21 | TOXCAST | ESOL | FREESOLV | LIPO |
|---|---|---|---|---|---|---|---|---|---|---|---|
| SMILES | 92.9±1.5 | 90.9±3.3 | 64.9±0.3 | 78.2±1.9 | **83.3±1.1** | 80.1±2.5 | 85.7±1.2 | 70.5±2.5 | 0.811± 0.109 | 1.623± 0.168 | 0.539± 0.017 |
| NMR$_{SPECTRUM}$ | 91.0±2.0 | **93.2±2.7** | **68.1±1.5** | **87.7±6.5** | 80.9±5.0 | 80.9±5.0 | 85.1±0.4 | **71.1±0.8** | 0.844± 0.123 | 2.417± 0.495 | 0.609± 0.031 |
| IMAGE | 93.1±2.4 | 92.9±1.8 | 65.3±1.5 | 86.2±6.5 | 82.3±0.6 | 78.7±1.7 | **86.0±1.0** | 71.0±1.6 | **0.761± 0.068** | 1.648± 0.045 | **0.537± 0.005** |
| FINGERPRINT | 92.9±2.3 | 91.7±3.6 | 65.6±0.7 | 87.5±6.0 | 81.2±2.5 | **82.9±3.1** | 85.3±1.3 | 70.0±1.4 | 0.808± 0.071 | **1.437± 0.134** | 0.565± 0.017 |
| NMR$_{PEAK}$ | **93.4±2.7** | 89.3±1.7 | 62.8±2.1 | 86.1±5.4 | 82.1±0.4 | 75.4±5.2 | 84.9±1.0 | 70.6±0.8 | 0.924±0.083 | 1.707±0.126 | 0.587±0.021 |
| AVERAGE | 92.8±1.9 | 91.4±2.7 | 65.3±2.0 | 85.0±5.7 | 81.8±2.2 | 79.4±4.0 | 85.4±0.9 | 70.6±1.3 | 0.830±0.094 | 1.766±0.394 | 0.586±0.048 |
| NO PRE-TRAINING | 91.9±3.0 | 85.2±0.6 | 57.0±0.7 | 90.6±0.6 | 77.1±0.5 | 78.6±1.4 | 75.9±0.7 | 63.7±0.2 | 1.050±0.008 | 2.082±0.082 | 0.683±0.016 |

All of the similarity calculation from the modalities above are listed in Appendix C.2.

### 4.1.2 PRE-TRAINING

NMRShiftDB-2 (Landrum, 2006) is a comprehensive database dedicated to nuclear magnetic resonance (NMR) chemical shift data, providing researchers with an extensive collection of expert-annotated NMR data for various organic compounds with molecular structures (SMILES). There are around 25,000 molecules used for pre-training and no overlap with downstream task datasets. And molecular images and graphs are generated via RDkit (RDK).

### 4.1.3 DOWNSTREAM TASKS

For Downstream tasks, our model was trained on 11 drug discovery-related benchmarks sourced from MoleculeNet (Wu et al., 2018a). Eight of these benchmarks were designated for classification downstream tasks, including BBBP, BACE, SIDER, CLINTOX, HIV, MUV, TOX21, and ToxCast, while three were allocated for regression tasks, namely ESOL, Freesolv, and Lipo. The datasets were divided into train/validation/test sets using a ratio of 80%:10%:10%, accomplished through the scaffold splitter (Halgren, 1996; Landrum, 2006) from Chemprop (Yang et al., 2019; Heid et al., 2023), like previous works. The scaffold splitter categorizes molecular data based on substructures, ensuring diverse structures in each set. Molecules are partitioned into bins, with those exceeding half of the test set size assigned to training, promoting scaffold diversity in validation and test sets. Remaining bins are randomly allocated until reaching the desired set sizes, creating multiple scaffold splits for comprehensive evaluation.

## 4.2 RESULTS

### 4.2.1 THE EFFECTIVENESS OF PRE-TRAINING

We first illustrate the impact of pre-training initialization on performance. As shown in Table 1, the average performance of pre-trained models outperform the non-pre-trained model in all tasks except for Clintox. The results of various downstream tasks indicate that different tasks may prefer different modalities. Notably, the model pre-trained with the NMR modality achieves the highest performance across three classification tasks. Similarly, the model pre-trained with the Image modality excels in three tasks, two of which are regression tasks related to solubility, aligning with findings from prior literature (Xu et al., 2023a). Additionally, the model pre-trained with The fingerprint method achieves the best performance in two tasks, including MUV, which has the largest dataset.

### 4.2.2 OVERALL PERFORMANCE OF MMFRL

As shown in Table 2 and Table 3, MMFRL demonstrates superior performance compared to all baseline models and the average performance of DMPNN pretrained with extra modalities across all 11 tasks evaluated in MoleculeNet. This robust performance highlights the effectiveness of our approach in leveraging multimodal data. In particular, while individual models pre-trained on other modalities for ClinTox fail to outperform the No-pretraining model (DMPNN), the fusion of these

Table 2: Overall performances (ROC-AUC) on classification downstream tasks. The best results are denoted in bold, and the second-best are indicated with underlining. For early fusion of MMFRL, all the predefined weight of each modality are 0.2. (Note: N-Gram is highly time-consuming on ToxCast.)

| DATA SET | BBBP | BACE | SIDER | CLINTOX | HIV | MUV | TOX21 | TOXCAST |
|---|---|---|---|---|---|---|---|---|
| ATTENTIVEFP | 64.3±1.8 | 78.4±2.2 | 60.6±3.2 | 84.7±0.3 | 75.7±1.4 | 76.6±1.5 | 76.1±0.5 | 63.7±0.2 |
| DMPNN | 91.9±3.0 | 85.2±0.6 | 57.0±0.7 | 90.6±0.6 | 77.1±0.5 | 78.6±1.4 | 75.9±0.7 | 63.7±0.2 |
| N-GRAM | 91.2±0.3 | 79.1±1.3 | 63.2±0.5 | 87.5±2.7 | 78.7±0.4 | 76.9±0.7 | 76.9±2.7 | - |
| GEM | 72.4±0.4 | 85.6±1.1 | **67.2±0.4** | 90.1±1.3 | 80.6±0.9 | 81.7±0.5 | 78.1±0.1 | 69.2±0.4 |
| UNI-MOL | 72.9±0.6 | 85.7±0.2 | 65.9±1.3 | 91.9±1.8 | 80.8±0.3 | 82.1±1.3 | 79.6±0.5 | 69.6±0.1 |
| INFOGRAPH | 69.2±0.8 | 73.9±2.5 | 59.2±0.2 | 75.1±5.0 | 74.5±1.8 | 74.0±1.5 | 73.0±0.7 | 62.0±0.3 |
| GRAPHCL | 67.5±3.3 | 68.7±7.8 | 60.1±1.3 | 78.9±4.2 | 75.0±0.4 | 77.1±1.3 | 75.0±0.3 | 62.8±0.2 |
| MOLCLR | 73.3±1.0 | 82.8±0.7 | 61.2±3.6 | 89.8±2.7 | 77.4±0.6 | 78.9±2.3 | 74.1±5.3 | 65.9±2.1 |
| MOLCLR$_{CMPNN}$ | 72.4±0.7 | 85.0±2.4 | 59.7±3.4 | 88.0±4.0 | 77.8±5.5 | 74.5±2.1 | 78.4±2.6 | 69.1±1.2 |
| GRAPHMVP | 72.4±1.6 | 81.2±9.0 | 63.9±1.2 | 79.1±2.8 | 77.0±1.2 | 77.7±6.0 | 75.9±5.0 | 63.1±0.4 |
| UNIMODALITY$_{avg}$ | 92.8±1.9 | 91.4±2.7 | 65.3±2.0 | 85.0±5.7 | 81.8±2.2 | 79.4±4.0 | **85.4±0.9** | _70.6±1.3_ |
| MMFRL$_{early}$ | 91.6±5.0 | _94.3±2.4_ | _66.4±1.9_ | 85.3±6.8 | 82.0±2.4 | 80.6±3.2 | 85.2±0.2 | 69.8±1.1 |
| MMFRL$_{intermediate}$ | **95.4±0.7** | **95.1±1.0** | 64.3±1.2 | **93.4±1.1** | 81.2±1.3 | **83.5±1.6** | 85.1±0.1 | **71.9±1.1** |
| MMFRL$_{late}$ | _94.7±0.6_ | 91.6±2.6 | 64.2±1.2 | 87.0±0.4 | **82.9±0.2** | 82.1±1.7 | 77.7±0.5 | 70.2±0.3 |

Table 3: Overall performances (RMSE) on regression downstream tasks. The best results are denoted in bold, and the second-best are indicated with underlining. For early fusion of MMFRL, all the predefined weight of each modality are 0.2.

| Data Set | ESOL | FreeSolv | Lipo |
|---|---|---|---|
| AttentiveFP | 0.877±0.029 | 2.073±0.183 | 0.721±0.001 |
| DMPNN | 1.050±0.008 | 2.082±0.082 | 0.683±0.016 |
| N-Gram$_{RF}$ | 1.074±0.107 | 2.688±0.085 | 0.812±0.028 |
| N-Gram$_{XGB}$ | 1.083±0.082 | 5.061±0.744 | 2.072±0.030 |
| GEM | 0.798±0.029 | 1.877±0.094 | 0.660±0.008 |
| Uni-Mol | 0.788±0.029 | 1.620±0.035 | 0.660±0.008 |
| MolCLR | 1.113±0.023 | 2.301±0.247 | 0.789±0.009 |
| MolCLR$_{CMPNN}$ | 0.911±0.082 | 2.021±0.133 | 0.875±0.003 |
| Unimodality$_{avg}$ | 0.924±0.083 | _1.707±0.126_ | 0.587±0.021 |
| MMFRL$_{early}$ | 1.037±0.170 | 2.093±0.090 | 0.607±0.034 |
| MMFRL$_{intermediate}$ | **0.730±0.019** | **1.465±0.096** | _0.552±0.014_ |
| MMFRL$_{late}$ | _0.763±0.035_ | 1.741±0.191 | **0.525±0.018** |

pre-trained models leads to improved performance. Besides, apart from Tox21 and Sider, the fusion models significantly enhances overall performance. In particular, the intermediate fusion model stands out by achieving the highest scores in seven distinct tasks, showcasing its ability to effectively combine features at a mid-level abstraction. the late fusion model achieves the top performance in two tasks. These results underscore the advantages of utilizing various fusion strategies in multimodal learning, further validating the efficacy of the MMFRL framework.

### 4.3 ANALYSIS OF THE FUSION EFFECT

#### 4.3.1 GENERAL COMPARISON AMONG VARIOUS WAYS OF FUSIONS

Early Fusion is employed during the pretraining phase and is easy to implement, as it aggregates information from different modalities directly. However, its primary limitation lies in the necessity for predefined weights assigned to each modality. These weights may not accurately reflect the relevance of each modality for the specific downstream tasks, potentially leading to suboptimal performance.

Intermediate Fusion is able to capture the interaction between modalities early in the fine-tuning process, allowing for a more dynamic integration of information. This method can be particularly beneficial when different modalities provide complementary information that enhances overall performance. If the modalities effectively compensate for one another's strengths and weaknesses, Intermediate Fusion may emerge as the most effective approach.

In contrast, Late Fusion enables each modality to be explored independently, maximizing the potential of individual modalities without interference from others. This separation allows for a thorough examination of each modality's contribution. When certain modalities dominate the performance metrics, Late Fusion can maximize on these strengths, ensuring that the most impactful information is utilized effectively. This approach is especially useful in scenarios where the dominance of specific modalities can be leveraged to enhance overall model performance.

### 4.3.2 Explainability of Learnt Representations

To demonstrate the interpretability of learnt representations of fusion, we present post-hoc analysis for two tasks, ESOL and Lipo, as demonstration. The results showcase learnt representations can capture task-specific patterns and offer valuable insights for molecular design.

**ESOL with Intermediate Fusion.** As presented in Table 3, the intermediate fusion method 3.2.2 exhibits superior performance on the ESOL regression task for predicting solubility. To further analyze this performance, we employed t-SNE to reduce the dimensionality of the molecule embeddings from 300 to 2, resulting in a heatmap visualized in Figure 2. The embeddings derived from individual modalities prior to fusion do not display a clear pattern, showing no smooth transition from low to high solubility. In contrast, the embeddings by intermediate fusion reveal a distinct and smooth transition in solubility values: molecules with similar solubility cluster together, forming a gradient that extends from the bottom left (indicating lower solubility) to the upper center (representing higher solubility). This trend underscores the effectiveness of the intermediate fusion approach in accurately capturing the quantitative structure-activity relationships for aqueous solubility.

Additionally, we examined the similarity between the respective embeddings prior to intermediate fusion and the resulting fused embedding, as depicted in Figure 3. Our analysis indicates that the embeddings from each modality exhibit low similarity with the intermediate-fused representation. This observation suggests that the modalities complement each other, collectively enhancing the resulting representation of the intermediate-fused embedding.

**Lipo with Late Fusion.** As detailed in Table 3, the Late Fusion method (described in Section 3.2.3) demonstrates superior performance on the Lipo regression task for predicting solubility in fats, oils, lipids, and non-polar solvents. According to Equation 11, the final prediction is determined by the respective coefficients ($w_i$) and predictions ($p_i$) from each modality.

In Figure 4, we present the distribution of values for the coefficients, predictions, and their products for each modality. Notably, the SMILES and Image modalities exhibit a broad range of values, suggesting their potential for significant contributions to the final predictions. This observation aligns with the strong performance achieved when pretraining using either of these two modalities, as shown in Table 1. In contrast, the $NMR_{Peak}$ values display a narrower range, indicating its role as a modifier for finer adjustments in the predictions. Furthermore, we observe that the contributions from $NMR_{Spectrum}$ and Fingerprint modalities are minimal, with their corresponding values approaching zero. This outcome highlights the advantages of the Late Fusion approach in effectively identifying and leveraging dominant modalities, thereby optimizing the overall predictive performance.

## 5 Related work

**Contrastive Learning on Molecular Graphs.** The primary focus within the domain of contrastive learning applied to molecular graphs centers on 2D-2D graphs comparisons. Noteworthy representative examples: InfoGraph (Sun et al., 2019) maximizes the mutual information between the representations of the graph and its substructures to guide the molecular representation learning; GraphCL (You et al., 2020), MoCL (Sun et al., 2021), and MolCLR (Wang et al., 2022b) employs graph augmentation techniques to construct positive pairs; MoLR (Wang et al., 2022a) establishes positive pairs with reactant-product relationships. In addition to 2D-2D graph contrastive learning, there are also noteworthy efforts exploring 2D-3D and 3D-3D contrastive learning in the field. 3DGCL (Moon et al., 2023) is 3D-3D contrastive learning model, establishing positive pairs with conformers from the same molecules. GraphMVP (Liu et al., 2022b), GeomGCL (Li et al., 2022), and 3D Informax (Stärk et al., 2022) proposes 2D–3D view contrastive learning approaches. To conclude, 2D-2D and 3D-3D comparisons are intra-modality contratsive leraning, as only one graph encoder is employed in these studies. And these approaches often focus on the motif and graph levels, leaving atom-level contrastive learning less explored.

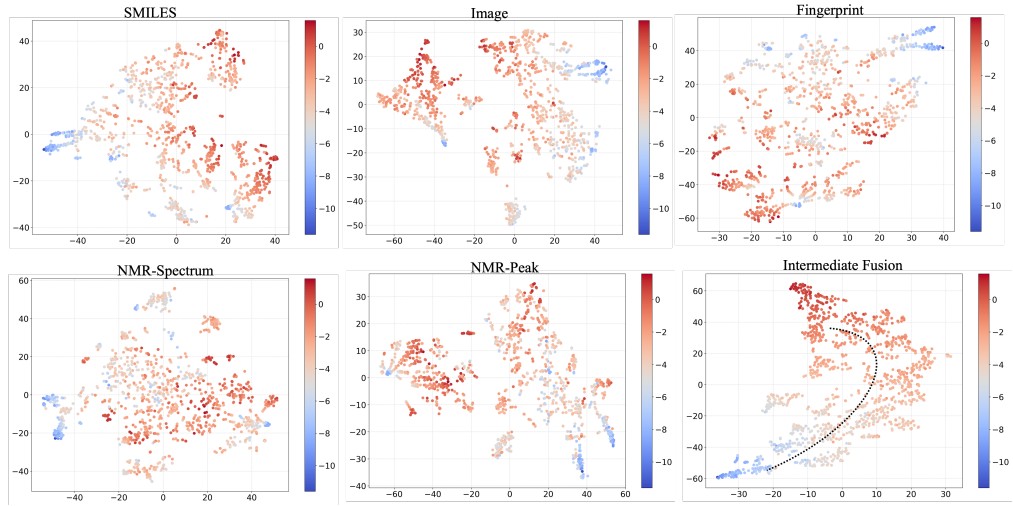

Figure 2: T-SNE visualization depicting the ESOL molecule embeddings for intermediate fusion in Section 3.2.2 alongside molecules within the highlighted region. Each point in the heatmap corresponds to the embeddings of respective molecules in ESOL, with color indicating solubility levels. Red denotes higher solubility, while blue indicates lower solubility. The embeddings derived from individual modalities prior to fusion do not display a clear pattern, the embeddings by intermediate fusion forms a gradient that extends from the bottom left (indicating lower solubility) to the upper center (representing higher solubility).

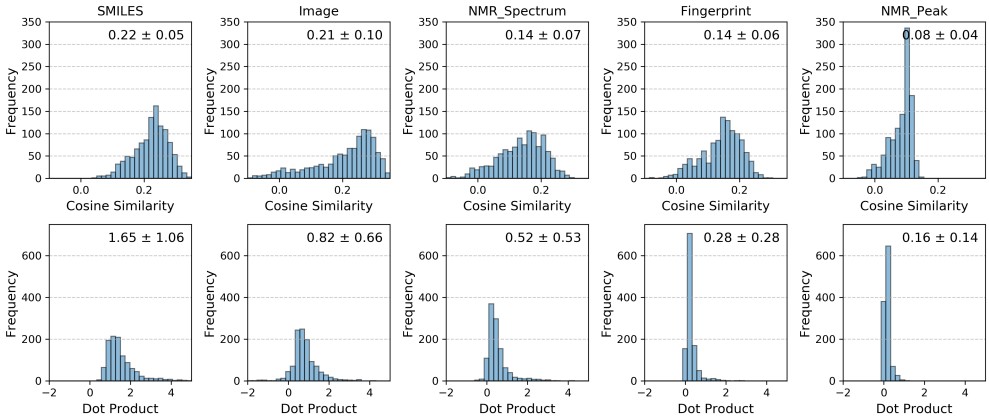

Figure 3: This figure shows the distribution of similarities between each modality and the intermediate fusion embedding for ESOL. In both Cosine Similarity and Dot Product, the embeddings from each modality exhibit low similarity with the intermediate-fused representation.

**Similarity Learning.** Instance-wise discrimination, a crucial facet of similarity learning, involves evaluating the similarity between instances directly based on their latent representations or features (Wu et al., 2018b). Naive instance-wise discrimination relies on pairwise similarity, leading to the development of contrastive loss (Hadsell et al., 2006). Although there are improved loss functions such as triplet loss (Hoffer & Ailon, 2015), quadruplet loss (Law et al., 2013), lifted structure loss (Oh Song et al., 2016), N-pairs loss (Sohn, 2016), and angular loss (Wang et al., 2017), these methods still fall short in thoroughly capturing relationships among multiple instances simultaneously (Wang et al., 2019). To address this limitation, a joint multi-similarity loss has been proposed, incorporating pair weighting for each pair to enhance instance-wise discrimination (Wang et al., 2019; Zhang et al., 2021). Notably, it is crucial to emphasize that employing these pair weightings requires the manual

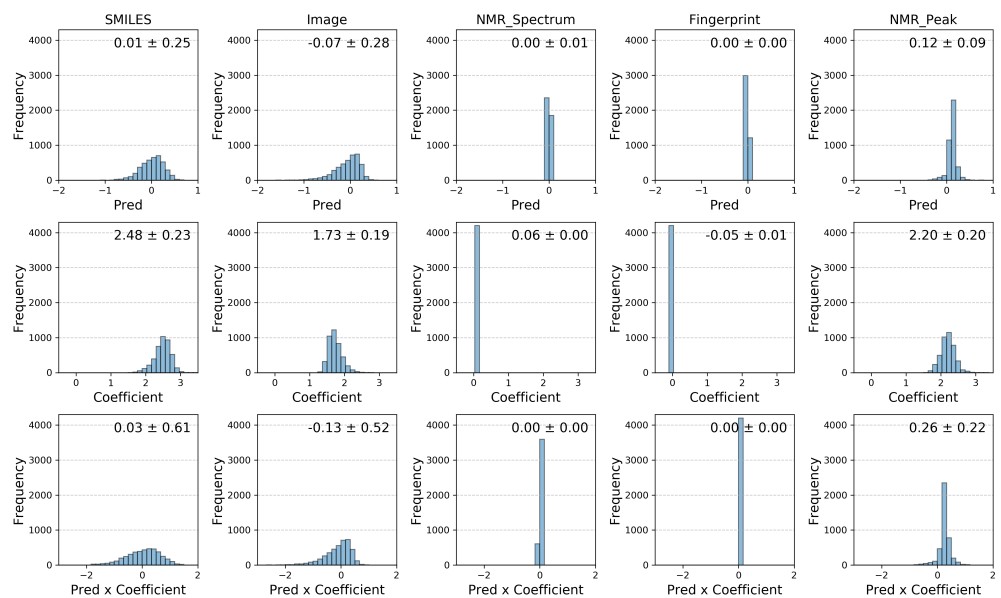

Figure 4: Lipo late fusion contribution analysis reveals that the three primary contributors are SMILES, image, and NMR$_{peak}$. In contrast, NMR$_{spectrum}$ and fingerprint exhibit negligible contributions.

categorization of negative and positive pairs, as distinct weights are assigned to losses based on their categories.

## 6 DISCUSSION

In summary, we introduce a novel relational learning metric for molecular graph representation that enhances the understanding of inter-instance relationships by capturing both local and global contexts. This is the first implementation of such a generalized metric in molecular graphs.Our method transforms pairwise self-similarity into relative similarity through a weighting function, allowing for complex relational insights. This metric is integrated into a multimodal representation, improving performance by utilizing modalities not directly accessible during fine-tuning. Empirical results show that our approach, MMFRL, excels in various molecular property prediction tasks. We also demonstrate detailed study about the explainability of the learned representations, offering valuable insights for drug molecule design. Despite these accomplishments, further exploration is needed to achieve more effective integration of graph- and node-level similarities. Looking ahead, we are enthusiastic about the prospect of applying our model to additional fields, such as social science, thereby broadening its applicability and impact.

## ACCESSIBILITY

The code and dataset will be made available upon the date of publication.

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

# Appendix

## A MULTI-SIMILARITY & CONTRASTIVE LEARNING

### A.1 MULTI-SIMILARITIES IN CONTRASTIVE LEARNING

Two distinct types of similarities, as illustrated in Appendix Figure A.1, can be identified: *self-similarity* (the pairwise similarity between two objects, typically defined through cosine similarity) and *relative similarity* (distinctions in self-similarity with other pairs) (Wang et al., 2019).

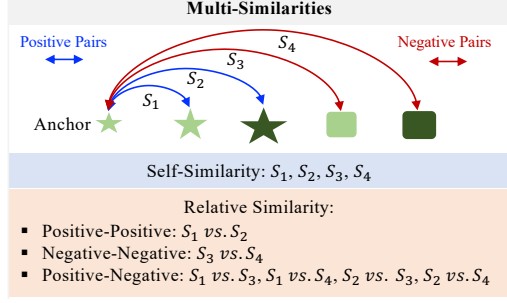

Figure A.1: Illustration of Different Types of Similarities.

### A.2 CURRENT MOLECULAR GRAPH CONTRASTIVE LEARNING APPROACHES

In current molecular graph contrastive learning approaches, positive pairs are commonly formed through either *data augmentation* (Sun et al., 2021; You et al., 2020), employing techniques such as node deletion, edge perturbation, subgraph extraction, attribute masking, and subgraph substitution, or *domain knowledge*, as demonstrated by reactant-product pairing (Wang et al., 2022a) or conformer grouping (Moon et al., 2023).

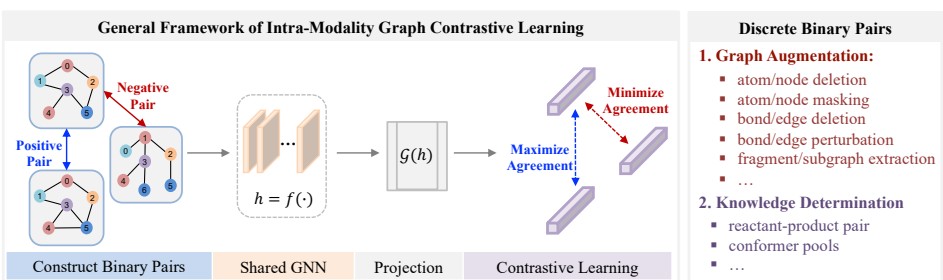

Figure A.2: General framework of Intra-Modality Graph Contrastive Learning. It relies on definition of positive and negative pairs.

## B SUPPLEMENTARY PROOF

### B.1 REVISITING THEOREM OF CONVERGENT SIMILARITY LEARNING

Let $\mathcal{S}$ be a set of instances with size $|\mathcal{S}|$, and let $\mathcal{P}$ represent the tunable latent representations of instances in $\mathcal{S}$ such that $|\mathcal{P}| = |\mathcal{S}|$. For any two instances $i, j \in \mathcal{S}$, their latent representations are denoted by $\mathcal{P}_i$ and $\mathcal{P}_j$, respectively. Let $t_{i,j}$ represent the target similarity between instances $i$ and $j$ in a given domain, and $d_{i,j}$ be the similarity between $\mathcal{P}_i$ and $\mathcal{P}_j$ in the latent space.

**Theorem B.1** (Theorem of Convergent Similarity learning). *Given $t_{i,j}$ is non-negative and $\{t_{i,j}\}$ satisfies the constraint $\sum_{j=1}^{|\mathcal{S}|} t_{i,j} = 1$, consider the loss function for an instance $i$ defined as follows:*

$$L(i) = -\sum_{j=1}^{|\mathcal{S}|} t_{i,j} \log \left( \frac{e^{d_{i,j}}}{\sum_{k=1}^{|\mathcal{S}|} e^{d_{i,k}}} \right) \tag{B.1}$$

*then when it reaches ideal optimum, the relationship between $t_{i,j}$ and $d_{i,j}$ satisfies:*

$$softmax(d_{i,j}) = t_{i,j} \tag{B.2}$$

*Proof.* In order to optimize the loss $L(i)$, we need to set the following partial derivative to be 0 for each $d_{i,j}$ with $1 \leq j \leq |\mathcal{M}|$. Here are the detailed steps:

$$\frac{\partial L(i)}{\partial d_{i,j}} = \frac{\partial}{\partial d_{i,j}} \underbrace{\left( -t_{i,j} \log \frac{e^{d_{i,j}}}{e^{d_{i,j}} + \sum_{k \neq j} e^{d_{i,k}}} \right)}_{\text{When the numerator includes } e^{d_{i,j}}} + \frac{\partial}{\partial d_{i,j}} \underbrace{\left( \sum_{k \neq j} -t_{i,k} \log \frac{e^{d_{i,k}}}{e^{d_{i,j}} + \sum_{k \neq j} e^{d_{i,k}}} \right)}_{\text{When the numerator does not include } e^{d_{i,j}}}$$

$$= -(t_{i,j} - t_{i,j} \cdot \text{softmax}(d_{i,j})) - \sum_{k \neq j} t_{i,k} \cdot \text{softmax}(d_{i,j})$$

$$= -\left( t_{i,j} - \left( t_{i,j} + \sum_{k \neq j} t_{i,k} \right) \cdot \text{softmax}(d_{i,j}) \right)$$

Since $\sum_{l=1}^{|\mathcal{M}|} t_{i,l} = 1$, we can further simplify it as

$$\frac{\partial L(i)}{\partial d_{i,j}} = -(t_{i,j} - \text{softmax}(d_{i,j}))$$

In order to optimize, we need to see the above partial derivative to be 0:

$$\frac{\partial L(i)}{\partial d_{i,j}} = -(t_{i,j} - \text{softmax}(d_{i,j})) = 0$$

In addition, the corresponding second partial derivative denoted as $\frac{\partial L(i)}{\partial d_{i,j}^2}$ manifests as follows:

$$\frac{\partial L(i)}{\partial d_{i,j}^2} = \text{softmax}(d_{i,j})(1 - \text{softmax}(d_{i,j}))$$

As $\text{softmax}(d_{i,j})$ takes values within the open interval (0,1), it follows that $\frac{\partial L(i)}{\partial d_{i,j}^2}$ is always positive. Consequently, the global optimum is global minimum.
Furthermore, when it comes to optimum:

$$t_{i,j} = \text{softmax}(d_{i,j})$$

$$d_{i,j} = \log(t_{i,j}) + \log \left( \sum_{1 \leq l \leq |\mathcal{M}|} e^{d_{i,j}} \right)$$

It is easy to show that when it reaches optimum, $d_{i,j}$ is consistent with target similarity metric $t_{i,j}$. Without loss of generosity, suppose $t_{i,j} > t_{i,j'}$:

$$d_{i,j} - d_{i,j'} = \log(t_{i,j}) + \log \left( \sum_{1 \leq l \leq |\mathcal{M}|} e^{d_{il}} \right) - \left( \log(t_{i,j'}) + \log \left( \sum_{1 \leq l \leq |\mathcal{M}|} e^{d_{il}} \right) \right)$$

$$= \log(t_{i,j}) - \log(t_{i,j'})$$

$$= \log \left( \frac{t_{i,j}}{t_{i,j'}} \right) > 0$$

$\square$

### B.2 GUARANTEE OF SUM OF FUSED MULTIMODAL SIMILARITY

Given sets of uni-modal generalized similarity $\{t^R\}$ and $\sum w_{t^R} = 1$, the sum of fused multimodal similarity also equals 1, as demonstrated below:

$$\sum(t_{i,j}^R) = \sum\sum(w_R \cdot t_{i,j}^R)$$
$$= \sum(w_R \sum t_{i,j}^R)$$
$$= \sum w_R \cdot 1 = 1$$

## C REVISITING TARGET SIMILARITY SETTINGS

### C.1 ENCODERS & PACKAGES

To derive the target similarities, we need to reply on pre-trained encoders or well-defined packages as follows:

Table C.1: Encoders and packages used to produce self-similarities

| Unimodal | Representation | Encoder/Package | Pre-trained Source |
|---|---|---|---|
| Image | 2D image | CNN | Img2mol (Clevert et al., 2021) |
| SMILES | Sequence | Transformer | CReSS (Yang et al., 2021) |
| $^{13}$CNMR Spectrum | Sequence | 1D CNN | AutoEncoder (Costanti et al., 2023) |
| $^{13}$CNMR peak | Scalar | NMRShiftDB2 (Steinbeck et al., 2003) | N/A |
| Fingerprint | Sequence | RDKit (Landrum, 2006) | N/A |

### C.2 TARGET SIMILARITY AT GRAPH LEVEL

**Fingerprint.** The mathematical formula of fingerprint similarity, denoted as $S_{i,j}^F$, can be viewed as follows:

$$S_{i,j}^F = Tanimoto(A, B) = \frac{|A \cap B|}{|A \cup B|} \tag{C.1}$$

where $A$ and $B$ are sets of molecular fragments for molecule $i$ and $j$, and $|A \cap B|$ and $|A \cup B|$ denote the size of their intersection and union, respectively.

**Image.** The self-similarity for Image, denoted as $S_{i,j}^I$, can be defined as follows:

$$S_{i,j}^I = Cos(\mathcal{V}_i, \mathcal{V}_j) = \frac{\mathcal{V}_i \cdot \mathcal{V}_j^T}{\|\mathcal{V}_i\| \cdot \|\mathcal{V}_j\|} \tag{C.2}$$

where $\mathcal{V}_i, \mathcal{V}_j$ represents the embedding of Image for two given molecules.

**NMR Spectrum.** The self-similarity for NMR spectrum, denoted as $S_{i,j}^C$, can be defined as follows:

$$S_{i,j}^C = Cos(\mathcal{V}_i, \mathcal{V}_j) = \frac{\mathcal{V}_i \cdot \mathcal{V}_j^T}{\|\mathcal{V}_i\| \cdot \|\mathcal{V}_j\|} \tag{C.3}$$

where $\mathcal{V}_i, \mathcal{V}_j$ represents the embedding of NMR spectra for two given molecules.

**Smiles.** The self-similarity for Smiles, denoted as $S_{i,j}^S$, can be defined as follows:

$$S_{i,j}^S = Cos(\mathcal{V}_i, \mathcal{V}_j) = \frac{\mathcal{V}_i \cdot \mathcal{V}_j^T}{\|\mathcal{V}_i\| \cdot \|\mathcal{V}_j\|} \tag{C.4}$$

where $\mathcal{V}_i, \mathcal{V}_j$ represents the embedding of Smiles for two given molecules.

**NMR Peak** The similarity among nodes (atoms) is derived from the positions of their signal peaks on $^{13}$C NMR spectra, measured in parts per million (ppm). The ppm values are continuous, typically

ranging from 0 to 200 (see more introduction of ppm in Appendix C.3). The self-similarity of NMR peaks $S_{l,m}^P$ can be defined as following:

$$S_{l,m}^P = \frac{\tau_2}{|ppm_l - ppm_m| + \tau_1} \tag{C.5}$$

where $ppm_l$ and $ppm_m$ are the positions of NMR peaks for the $l^{th}$, $m^{th}$ Carbon atom, $\tau_1$ and $\tau_2$ are temperature hyper-parameter.

### C.3 A Brief Introduction to PPM for NMR Peak

In chemistry, $^{13}$C NMR stands out as a common technique for structural analysis by revealing molecular structures by elucidating the chemical environments of carbon atoms and their magnetic responses to external fields (Gerothanassis et al., 2002; Lambert et al., 2019). It quantifies these features in parts per million (ppm) relative to a reference compound, such as tetramethylsilane (TMS), thereby simplifying comparisons across experiments. As a result, the continuous peak positions, measured in parts per million (ppm), offer a robust knowledge span—a natural ordering metric that can be employed to derive measures of similarity (Xu et al., 2023b).

### C.4 Configuration of Early Fusion

A simple linear combination is used to formulate the multimodal relational similarity $t_{i,j}^M$ between the $i^{th}$ and $j^{th}$ molecules, represented as as follows:

$$t_{i,j}^M = w_{SM} \cdot t_{i,j}^{SM} + w_C \cdot t_{i,j}^C + w_I \cdot t_{i,j}^I + w_F \cdot t_{i,j}^F + w_F \cdot t_{i,j}^F + w_P \cdot t_{i,j}^P \tag{C.6}$$

where $t_{i,j}^{SM}$ denotes the similarity based on SMILES, $t_{i,j}^C$ denotes the similarity with respect to $^{13}$C NMR spectrum, $t_{i,j}^I$ denotes the similarity regarding images, $F$ denotes the similarity based on fingerprints, and $P$ denotes the similarity based on fingerprints. $w_{SM}$, $w_C$, $w_I$, and $w_F$ are the pre-defined weights for their respective similarity, and $w_{SM} + w_C + w_I + w_F + w_P = 1$.

## D Experimental Settings

### D.1 Pre-Training Setting

During pretraining, we utilized an Adam optimizer with a learning rate set to 0.001, spanning 200 epochs and employing a batch size of 256. The model was trained on around 25,000 data points. The NMR data were experimental data, extracted from NMRShiftDB2 (Steinbeck et al., 2003). Other chemical modalities, such as images, fingerprints and graphs, were produced from SMILES by RDKit (Landrum, 2006).

### D.2 Fine-Tuning Setting

#### D.2.1 Datasets

For fine-tuning, our model was trained on 11 drug discovery-related benchmarks sourced from MoleculeNet (Wu et al., 2018a). Eight of these benchmarks were designated for classification downstream tasks, including BBBP, BACE, SIDER, CLINTOX, HIV, MUV, TOX21, and ToxCast, while three were allocated for regression tasks, namely ESOL, Freesolv, and Lipo. The datasets were divided into train/validation/test sets using a ratio of 80%:10%:10%, accomplished through the scaffold splitter (Halgren, 1996; Landrum, 2006) from Chemprop (Yang et al., 2019; Heid et al., 2023), like previous works. The scaffold splitter categorizes molecular data based on substructures, ensuring diverse structures in each set. Molecules are partitioned into bins, with those exceeding half of the test set size assigned to training, promoting scaffold diversity in validation and test sets. Remaining bins are randomly allocated until reaching the desired set sizes, creating multiple scaffold splits for comprehensive evaluation.

### D.2.2 BASELINES

We systematically compared MMFRL's performance with various state-of-the-art baseline models across different categories. In the realm of supervised models, AttentiveFP (Xiong et al., 2019) and DMPNN (Yang et al., 2019) stand out by leveraging graph attention networks and node-edge interactive message passing, respectively. The unsupervised learning method N-Gram (Liu et al., 2019) employs graph embeddings and short walks for graph representation. Predictive self-supervised learning methods, such as GEM (Fang et al., 2022) and Uni-Mol (Zhou et al., 2023), are specifically designed for predicting molecular geometric information. Moreover, our evaluation encompasses a range of contrastive learning methods, namely InfoGraph (Sun et al., 2019), GraphCL (You et al., 2020), MolCLR (Wang et al., 2022b), and GraphMVP (Liu et al., 2022b), all serving as essential baselines. The baseline results are collected from recent works (Fang et al., 2022; Zhou et al., 2023; Moon et al., 2023; Fang et al., 2023).

### D.2.3 EVALUATION

To assess the effectiveness of our fine-tuned model, we measure the ROC-AUC for classification downstream tasks, and the root mean squared error (RMSE) metric for regression tasks. In order to ensure a fair and robust comparisons, we conduct three independent runs using three different random seeds for scaffold splitting across all datasets. The reported performance metrics are then averaged across these runs, and the standard deviation is computed as prior works.

