# OpenReview forum: "Multimodal Fusion with Relational Learning for Molecular Property Prediction"
_ICLR.cc/2025/Conference — ICLR 2025 Conference Withdrawn Submission_

### Official Review · Reviewer_Z9Gs · 2024-10-17

**Soundness:** 2
**Presentation:** 2
**Contribution:** 1
**Rating:** 3
**Confidence:** 3

**Summary:**

This paper addresses challenges in graph-based molecular representation learning. They propose MMFRL, which enhances embedding initialization through multimodal pre-training and investigates fusion at various stages. MMFRL aims to improve downstream performance on molecular property prediction tasks by leveraging relational learning and multimodal fusion.

**Strengths:**

1. The paper introduces a relational learning metric that moves beyond binary metrics, allowing a continuous perspective on molecular relationships.

3. The method fuses on different stages and adapts well to various scenarios. The intermediate fusion method stands out in multiple tasks, as it allows a more dynamic integration of information from different modalities.

3. The model achieves superior performance over several baselines on molecular property prediction tasks across multiple datasets from MoleculeNet.

**Weaknesses:**

1. In terms of multimodal fusion, previous works [1,2] have introduced methods for multimodal molecular graph-text pre-training, with one of them [2] also incorporating SMILES strings. These models demonstrate strong performance not only on molecular property prediction tasks but also on a range of multimodal downstream tasks. The authors should also compare their approach with these existing models for a more comprehensive evaluation.

2. Why do models with different stage fusions perform differently across datasets? The authors need to explain this.

3. typos: line 204: themto

**Questions:**

1. What's the difference between Relation Learning and Contrastive Learning?

2. The authors should also compare their approach with [1,2] for a more comprehensive evaluation.

3. How do the modified relational learning metric help to capture both localized and global relationships?

[1] Multi-modal Molecule Structure-text Model for Text-based Retrieval and Editing

[2] MolCA: Molecular Graph-Language Modeling with Cross-Modal Projector and Uni-Modal Adapter

---

### Official Review · Reviewer_6WqF · 2024-10-23

**Soundness:** 2
**Presentation:** 3
**Contribution:** 2
**Rating:** 3
**Confidence:** 4

**Summary:**

This paper introduces a new metric for measuring inter-molecular relationships and applies this metric to molecular multimodal fusion. In the context of molecular multimodal fusion, the paper integrates various intra-molecular modalities and conducts ablation studies on the fusion at different stages and the contributions of different modalities. The effectiveness of the proposed method is validated through downstream tests on molecular property prediction tasks using the MoleculeNet dataset.

**Strengths:**

1. The writing of the paper is clear and comprehensible.
2. The visualization design in Figure 2 is meaningful and clear, effectively illustrating the advantages of multimodal fusion.

**Weaknesses:**

1. Regarding the key contribution emphasized in the paper—the measurement of inter-molecular relationships—it is noted that the smooth contrastive learning metric has been previously proposed in various studies. The paper appears to merely transfer this metric to the molecular domain for application.
2. Concerning the work on multimodal fusion, the focus of the paper is on exploring the optimal fusion stage. However, it seems that most existing fusion works also perform fusion at this identified optimal stage. While conducting an ablation study on this empirical approach is somewhat meaningful, it appears somewhat limited as a primary contribution of the paper.
3. The downstream tasks conducted in the paper are rather limited, which does not provide compelling evidence for the effectiveness of the proposed method. The experiments are solely tested on the MoleculeNet dataset, which generally has a small sample size, leading to a degree of randomness in the results. Given that the title of the paper specifically addresses molecular property prediction, it is also essential to test on classic large-scale molecular property prediction datasets, such as PCQM4Mv2 and QM9.
4. The design of the figures and the structure of the paper require improvement. The figures are not presented in a reasonable format, and the font is somewhat unclear, making them difficult to read. Additionally, the overall structure of the paper appears somewhat disorganized, particularly in the experimental section, where a more logical design could be adopted.

**Questions:**

1. Given that the paper claims that the proposed method can contribute to drug design, I am particularly interested in its performance on large molecule-size datasets with significant practical value, such as the LBA dataset.
2. I am not very familiar with the concept of relational learning mentioned in the paper. Is relational learning a specialized research area? What is its connection to contrastive learning?
3. I am curious about the generalization ability of the proposed method on out-of-distribution data, particularly when training and testing involve molecules with varying ranges of heavy atom counts. A common challenge in molecular property prediction is the poor generalization ability on out-of-distribution data. I wonder whether the integration of multiple intra-molecular modalities can alleviate this issue to some extent.

---

### Official Review · Reviewer_v7ZL · 2024-11-03

**Soundness:** 3
**Presentation:** 2
**Contribution:** 2
**Rating:** 3
**Confidence:** 4

**Summary:**

The paper proposes MMFRL (Multimodal Fusion with Relational Learning for Molecular Property Prediction), a new framework to address challenges in graph-based molecular representation learning for drug discovery and materials science. It enhances embedding initialization through multimodal pre-training with relational learning and systematically explores the effects of modality fusion at different stages (early, intermediate, late) to understand their benefits and limitations. Experiments on MoleculeNet benchmarks show that MMFRL outperforms existing methods and offers explainability that provides chemical insights, highlighting its potential for real-world applications in drug discovery.

**Strengths:**

1.	This paper addresses the important topic of pre-trained molecular representation learning. The authors comprehensively consider multiple modalities such as Fingerprint, SMILES, NMR, and Image, and systematically explore the effects of modality fusion at different stages (early, intermediate, late) to understand their benefits and limitations.
2.	The authors evaluate the model’s performance across various molecular property prediction datasets.

**Weaknesses:**

1.The paper states that while contrastive learning is often used to handle molecular relationships, its reliance on binary metrics is insufficient for capturing the complexity of these interactions. However, this research question is not novel in the context of molecular pretraining. Prior work, such as MoleBERT [1], has already discussed this issue in the context of molecular graphs and proposed corresponding solutions. Unfortunately, the authors did not reference this previous work.
2.The introduction of multimodal contrastive learning to address the limitations of binary metrics has limited novelty.
3.The baselines used for comparison are not comprehensive. For example, in Table 2, there is no comparison with the current state-of-the-art model, Uni-Mol [2].
4.The authors only evaluated performance on molecular property prediction tasks, lacking assessments on other tasks such as molecule-molecule interaction prediction and molecule-target affinity prediction.
5. There is no code or checkpoint provided for reproducing the results.

[1] Mole-BERT: Rethinking Pre-training Graph Neural Networks for Molecules (ICLR 2023)
[2] Uni-Mol: A Universal 3D Molecular Representation Learning Framework (ICLR 2023)

**Questions:**

See the weakness above

---

### Official Review · Reviewer_H7a7 · 2024-11-04

**Soundness:** 2
**Presentation:** 2
**Contribution:** 2
**Rating:** 5
**Confidence:** 4

**Summary:**

The paper introduces Multimodal Fusion with Relational Learning (MMFRL), a framework that enhances molecular property prediction through multimodal fusion and relational learning. The authors applied relational learning on molecular prediction by modifying relational learning loss with a continuous similarity metric, to capture nuanced differences in molecular distances through relative positioning among molecules. By utilizing multimodal learning with a pretrained encoder for each modality, they aimed to obtain a unified molecular representation that incorporates information from multiple modalities. To fuse information from these modalities, they proposed three distinct fusion strategies, each demonstrating unique characteristics in representation. This approach led to varied performance patterns across 11 downstream tasks in MoleculeNet dataset, with MMFRL achieving superior performance.

**Strengths:**

-	The method effectively employs a continuous loss function for multimodal relational learning in molecular property prediction, delivering promising results.

-	The method introduces three distinct multimodal fusion approaches – early, intermediate, and late stage – offering alternative strategies for integrating different modalities and illustrating the strengths and weakness of each.

**Weaknesses:**

Major comments

-	While the idea of modifying the loss function for efficient relational learning sounds novel, the explanation on the specific modification of the loss function does not effectively support the novelty of the proposed method, as the objective function is a rather standard one widely used in the community.
-	Although the paper presents three approaches to multimodal fusion - early, intermediate, and late-stage fusion - it is not clear which of the three stage fusion strategies can be effectively used for certain tasks. Clearly demonstrating for which tasks each fusion strategy performs better would strengthen the work. For example, it remains unclear to users when would specific modality dominates the learning process, a case where the authors suggest late fusion is more effective.

Minor comments

-	The case studies are intriguing, but some results appear inconsistent. For example, while the analysis on the fusion effect suggests that late fusion performs best when one modality dominates, the case study on the Lipo regression task implies the multiple modalities complement each other for better results.
-	Ablation result regarding pretrained graph encoder does not seem to need to be placed as first result.
-	Additional analysis only performed on regression tasks.
-	Typos : Section 3.2.1 1st line: ‘themto’,  Section 4.2.2 5th line: ‘No-pretraining’

**Questions:**

-	Does removing a key modality result in a significantly larger performance drop compared to removing other modalities? Ablation studies seem to be needed to better demonstrate claim.
-	The paper suggests that lower cosine similarity between each modality’s embeddings and the fused embeddings indicates a complementary effect. Could additional information clarify this claim, such as comparing similarity in other scenarios?

---

### Note · Authors · 2025-04-09

I have read and agree with the venue's withdrawal policy on behalf of myself and my co-authors.

---

### Meta-Review · Area_Chair_DAHq · 2024-12-17

**Metareview:**

The paper introduces a method for relational learning and multimodal fusion in molecular property prediction. However, reviewers raised significant concerns regarding the novelty, experimental design, and completeness of the work. While the modification to the loss function is presented as novel, it does not sufficiently distinguish itself from prior methods, such as MoleBERT. The multimodal fusion strategies are discussed, but the paper lacks clarity on when each fusion stage is most effective. Additionally, comparisons with important state-of-the-art models, like Uni-Mol, and the evaluation on diverse tasks and datasets are missing. The paper also lacks code and checkpoints for reproducibility. Furthermore, the experiments are limited, focusing primarily on small datasets like MoleculeNet, which may lead to inconsistent results. The paper's structure and presentation also need improvement. These issues led to a decision to reject the submission.

**Additional Comments On Reviewer Discussion:**

The authors did not provide a rebuttal.

---

### Decision · Program_Chairs · 2025-01-22

Reject